# Amazon Fund 10 Years Later: Lessons from the World's Largest REDD+ Program

**Juliano Correa** [1,*] , **Richard van der Hoff** [2,3] **and Raoni Rajão** [3]

1   Sanford School of Public Policy, Duke University, Durham, NC 27708, USA
2   Nijmegen School of Management, Radboud Universiteit, Nijmegen 6525 AJ, The Netherlands; richard.vanderhoff@gmail.com
3   Department of Production Engineering, Universidade Federal de Minas Gerais, Belo Horizonte, MG 30161-010, Brazil; rajao@ufmg.br
*   Correspondence: juliano.correa@duke.edu; Tel.: +1-919-904-6453

**Abstract:** Results-Based Funding (RBF) for Reducing Emissions from Deforestation and Forest Degradation (REDD+) has become an important instrument for channeling financial resources to forest conservation activities. At the same time, much literature on conservation funding is ambiguous about the effectiveness of existing RBF schemes. Many effectiveness evaluations follow a simplified version of the principal-agent model, but in practice, the relation between aid providers and funding recipients is much more complex. As a consequence, intermediary steps of conservation funding are often not accounted for in effectiveness studies. This research paper aims to provide a nuanced understanding of conservation funding by analyzing the allocation of financial resources for one of the largest RBF schemes for REDD+ in the world: the Brazilian Amazon Fund. As part of this analysis, this study has built a dataset of information, with unprecedented detail, on Amazon Fund projects, in order to accurately reconstruct the allocation of financial resources across different stakeholders (i.e., governments, NGOs, research institutions), geographies, and activities. The results show that that the distribution of resources of the Amazon Fund lack a clear strategy that could maximize the results of the fund in terms of deforestation reduction. First, there are evidences that in some cases governmental organizations lack financial additionality for their projects, which renders the growing share of funding to this type of stakeholder particularly worrisome. Second, the Amazon Fund allocations did also not systematically have privileged the municipalities that showed the recent highest deforestation rates. rom the 10 municipalities with the higher deforestation rates in 2017, only 2 are amongst the top 100 receiving per/Ha considering the 775 municipalities from Legal Amazon. Third, the allocation of the financial resources from the Amazon Fund reflects the support of different projects that adopt significantly diverging theories of change, many of which are not primarily concerned with attaining further deforestation reductions. These results reflect the current approach adopted by the Amazon Fund, that do not actively seek areas for intervention, but instead wait for project submissions from proponents. As a consequence, project owners exert much influence on to the type of activities that they support how deforestation reduction is expected to be attained. The article concludes that the Amazon Fund as well as other RBF programs, should evolve over time in order to develop a more targeted funding strategy to maximize the long-term impact in reducing emissions from deforestation.

**Keywords:** REDD+; Amazon Fund; Results-Based Funding; benefit distribution; resource allocation; climate change funding; effectiveness; forest conservation funding

## 1. Introduction

The international allocation of funds to activities intended to funding forest conservation—directly or indirectly—is said to be a "highly cost-effective way of reducing greenhouse gas emissions on climate change" [1]. Among many types of financial mechanisms for pursuing this approach, Results-Based Funding (RBF) for Reducing Emissions from Deforestation and Forest Degradation (REDD, or REDD+ for a broader suite of activities) has become an important instrument for channeling financial resources to forest conservation activities [2,3]. RBF can be defined as the "transfer of money or material goods conditional upon taking a measurable action or achieving a predetermined performance target" [4–7]. The success of RBF instruments for REDD+ stems from political controversies related to initial REDD+ proposals that favored offset-based markets [8]. Brazilian government, in particular, has been known to challenge the use of markets on the basis of sovereignty concerns [6,9]. Instead, Brazil created the Amazon Fund in 2008 in order to receive results-based payments for achievements in deforestation reductions [10], which plummeted between 2004 and 2012 [11–13]. Similar developments have also occurred in international forest governance debates, as the Green Climate Fund became the central financial instrument for REDD+ [14], testifing the growing prevalence of RBF approaches in forest governance. Despite this dominance, the effectiveness of RBF has been challenged by scholars [5,7,15–18], while others have showed that donor and receiving countries and stakeholders often disagree on how to best evaluate these schemes and distribute the resources [19,20].

This research paper aims to enhance the understanding of intermediary stages of RBF for forest conservation by reconstructing the allocation of financial resources from the Brazilian Amazon Fund to individual projects and analyzing the underlying rationales behind this allocation. Between 2008 and 2017, the Amazon Fund has received more than USD 1.2 billion in donations, committed USD 667.3 million for the financial support of 96 approved projects, and thereby represents the largest and most longstanding RBF initiatives in forest governance worldwide [10,13,21]. An analysis of financial resource allocation could, therefore, provide important lessons on the intermediary stages of RBF (as Amazon Fund) to REDD+ and other conservation purposes. Our analysis exposes the underlying intervention logic (or 'theory of change') adopted for redistributing financial resources, which is useful for identifying the main factors for successful or failing forest conservation funding. The remainder of this paper proceeds as follows. Section 2 reviews the literature on related resource allocations, including the theories of change, criteria for resource allocation, benefit-sharing mechanisms, and impacts. Section 3 then outlines our approach and Section 4 presents data about the distribution of Amazon Fund resources. Section 5 concludes with our main findings and their implications for impact and policy making.

## 2. Aid Effectiveness and the Complex Relations between Service Providers and Service Users

Deforestation reduction [17,22] is a relatively recent trend in the broader context of development aid that has usually targeted health, education, or biodiversity conservation [16,23]. Although using the same model, the literature generally refers to aid as funding for REDD+ initiatives, since the former seems to be charity while the last is close to the climate change concepts, where developed countries should fund initiatives of forest conservation to offset their historical emissions [2].

Although this aid could come in many forms, RBF has become an increasingly appealing approach due to its simplicity from both the donor and receiver sides. On the donor side, the payments are done based on the measurement of a result already achieved, reducing substantially the transactional risk. On the receiver side, RBF promises the transfer of resources with "no strings attached" as countries are able to decide how best to invest the payments. Since receiving countries would want to receive an increasing volume of resources, they would be incentivized to invest the RBF proceedings in a way that reduces deforestation the most. A closer look, however, reveals that many of the issues that have plagued REDD+ and development aid more in general are still present in RBF, namely: benefit distribution, intervention design, and effectiveness.

One of the key design choices around REDD+ programs concerns the definition of "who needs to be involved, whose interests are at stake, and the expected co-benefits and required safeguards" [19]. Moreover, their discussion of approaches to reducing tropical forest degradation highlights the importance of contextualizing local realities, responding to new knowledge and experience, and incorporating the full complexity of forest loss and degradation, among others [24,25]. Many scholars have highlighted the issues of equitable sharing of net benefits from REDD+ projects (e.g., [26,27]). For instance, Luttrell, Loft, Fernanda Gebara, Kweka, Brockhaus, Angelsen, and Sunderlin [27] distinguish a number of possible rationales for the distribution of REDD+ benefits. They have emphasized: (1) actors with legal rights; (2) actors achieving reductions in emissions; (3) low-emitting forest stewards; (4) actors incurring the costs of REDD+ implementation; (5) effective facilitators of REDD+ implementation; and (6) the poorest actors. They note great variation in how implementing countries apply these rationales, implying that this is a function of context, project design and the beneficiaries (see also [8]). Some scholars find that "equity can have significant positive feedback on program outcomes and legitimacy over the longer term" [26,28,29]. According to Vatn and Vedeld [30], market-based approaches were found to be the most problematic among governance structures, since they do not address equity. These observations suggest a theme of providing equal opportunities to stakeholders. Yet rigorous analysis, and even comprehensive evaluations of net benefits and their distribution, are scarce, in part because of the way decisions are made about distributions of resources within and across REDD+ projects [19].

Another key aspect of RBF is the choice, by the receiving country, of the interventions that will be supported by the program. [27,31]. Weatherley-Singh and Gupta [32], for example, find that REDD+ activities must directly target the drivers of deforestation, such as forest fires and illegal logging, as well as structural drivers, such as changes in land tenure and land-use planning. However, they argue that not all drivers are considered, as most schemes do not address cattle ranching, corruption, roadbuilding, and/or commodity demands, among others (see also [9,33]). As important as the choice of the type of intervention is, the definition of the territories that will be prioritized by REDD+. Wolosin, Breitfeller, and Schaap [10] show that the geographical distribution of REDD+ finance can be largely explained by priorities on tree cover, tree-cover loss, and carbon emissions at national (70%–94%) and subnational (58%–72%) levels, though institutional capacity and political commitments have also been influential. Other work highlights significant gaps for specific priority areas. Some scholars point to areas in the Amazon region facing high deforestation pressure that are important for emissions and biodiversity [33–35]. Other scholars argue for additional investments in the network of protected areas, given their importance to date in curbing deforestation and the risks from deforestation dynamics [36,37]. Still others argue that support should also consolidate pristine or intact or stable forests to ensure long-term conservation (e.g., [35]). While the majority of available literature strongly emphasizes improved protection of high-risk areas, at the least for prioritizing additional impacts in the short run, various goals play parts within comprehensive approaches to forest conservation.

Finally, different studies have pointed out that it is not clear that RBF leads to the efficient use of resources, as initially assumed. The proponents of RBF expected that, since receiving countries have a direct financial incentive to reduce deforestation, they would strive to support actions on the ground that contribute directly to that aim. However, a closer look suggests that that empirical evidence on the effectiveness of RBF schemes is either lacking or points to contradictory effects [5], a problem already well known in relation to development aid [38]. On the one hand, authors such as Restivo, Shandra, and Sommer [17] argue that more bilateral aid from the United States Agency for International Development (USAID) has a lowering effect on forest loss. On the other hand, studies such as Hermanrud and de Soysa [22] report that forest conservation funding from Norway's International Forest and Climate Initiative (NICFI), one of the largest aid initiatives in the world and the main donor to the Amazon Fund, has had no effect in halting forest area loss. In a similar way, Bare, Kauffman, and Miller [18], for example, argue that forest conservation funding in sub-Saharan

Africa "is not associated with reduced deforestation rates at the national scale" and even claim that short-term impacts had negative effects. Both studies have strong limitations, since they do not control for other drivers of deforestation, such as agricultural prices, and they assume that relatively small-scale programs (as a percentage of the country's Gross National Product), are going to show effects at national level [39]. Nevertheless, these studies show that there is a growing concern with the effectiveness of RBF, in general and NICFI in particular.

The problem with evaluating the effectiveness of RBF initiatives is that the relations between service users (aid providers) and service providers (aid users) are much more complex than a simplified reading of the principal-agent model found in the studies cited above. According to Paul [7], the contracted agency relationship is often one between the donor organization and a recipient organization or ministry, whereas results may come from other organizations that ultimately spend the financial resources from these donations but have no direct relation with the donor organization (i.e., non-contracted agency relation). In this respect, for example, the UN-REDD+ programme from the United Nations Development Program (UNDP) supports 94 projects in Cambodia, Sri Lanka, Panama, Paraguay, Democratic Republic of the Congo, and Nigeria. However, UNDP are directly related only to the governmental focal point of each country, relating only indirectly with the local beneficiary [40].

According to Van der Hoff, Rajão and Leroy [19], the indirect relations between financial donations, 'project performance', and deforestation rates underlie discursive tensions between donor and recipient countries. While formally all parties agree that RPF should be based solely on deforestation reductions already achieved, donors are also increasingly concerned with the lack of evidence of efficiency of funded projects in driving additional reductions, and in this way fueling a virtuous circle. These tensions and conflicts suggest that the intermediary processes of forest conservation funding are poorly understood, particularly with respect to how they affect aid effectiveness. Some authors have suggested that addressing these conflicts requires new approaches to aid effectiveness evaluations that account for the complex relations of RBF for REDD+, particularly the intermediary stages of forest conservation funding. This could imply, for instance, that transfers should be conditional upon desired results, as within well-implemented payments for ecosystem services (PES) approaches [28]. Such conditions could also require environmental additionality—that is, providing more ecosystem services than they would provide in the non-existence of such payments [41,42]. In addition, REDD+ should be 'financially additional', beyond already planned funding [43]. While attractive, the idea of adding specific demands of additionality to RBF goes against the simplicity and 'hands off' approach that made RBF popular in the first place. Furthermore, this approach would entail a return of many elements of the project-based model defined by Verified Carbon Standard among others, which have also proven to be highly problematic [44]

The growing body of literature presented above presents valuable insights on how RBF should be designed and presents some of its dilemmas and contradictory results. But while allot has been said about how large RBF programs should look like, until recently we lacked a strong record of largescale schemes to look back and draw lessons from concrete experiences. This study provides the first comprehensive analysis of the first decade of the Amazon Fund, the world largest REDD+ RBF program [45,46]. Our study aims to reveal the design choices adopted by the Fund by analyzing its resource distribution across beneficiaries, activities, and geographies. While this study does not provide a quantitative impact analysis of the fund, it allows us to understand how the allocation of financial resources corresponds with various REDD+ design choices, as reflected in the available literature on REDD+, and the extent to which this may affect its long-term effectiveness. From this, this study draws lessons that could be used to improve the Amazon Fund in Brazil and other large RBF programs.

## 3. Research Approach and Methodology

This research paper conceptualizes the Amazon Fund as an intermediary organization that links the forest conservation funding provided by donor organizations to the individual projects

(see Figure 1). Created in 2008, the Amazon Fund was the first large scale RBF program to be implemented. As such, the fund played an important role in shaping the discussions around REDD+ at the United Nations Framework Convention on Climate Change (UNFCCC). For this reason, the UNFCCC's Warsaw Framework for REDD+ adopted, to a large degree, the modus operandi pioneered by Brazil. Financial donations to the Amazon Fund mainly come from Norway's International Climate and Forest Initiative (NICFI) and the German Development Bank (KfW). The Amazon Fund consists of a steering committee (COFA), which is responsible for establishing allocation guidelines, and a technical committee (CTFA), which is responsible for approving results in terms of reducing emissions from deforestation. The managing organization of the Amazon Fund is the Brazilian Development Bank (BNDES) and is responsible for the approval (or rejection) of submitted project proposals according to predefined guidelines, as well as for the receipt and allocation of financial resources. Since 2015, BNDES has also become eligible to receive financial resources from the Green Climate Fund (decree 8.576/15), whereas other organizations like the government-owned bank Caixa Econômica Federal (CEF) and the Brazilian Biodiversity Fund (FUNBIO) may also become recipients. Financial resources are allocated to a wide variety of organizations. Federal government organizations include the Brazilian Agricultural Research Corporation (EMBRAPA), the Brazilian Institute for Space Research (INPE), the Brazilian Institute for the Environment and Renewable Natural Resources (IBAMA) and the National Police Force (FNSP). Non-governmental organizations also abound and include the Sustainable Amazon Foundation (FAS), the Amazon Institute for Human and Environment (IMAZON), Amazon Environmental Research Institute (IPAM), and The Nature Conservancy (TNC), among others. State government organizations are mostly represented by the environmental or agricultural secretariats of the nine Brazilian states in the Legal Amazon, while some state secretariats outside this region were also recipients. Finally, municipal government secretariats and federal universities were also supported financially by the Amazon Fund.

Understanding how forest conservation funding to the Amazon Fund contributes to the effective reduction of emissions from deforestation and forest degradation involves connecting the project activities (each with a specific shared benefit), geographies, and supported activities, to the overall objective of emissions reduction. The Amazon Fund already provides an annual report that divides the funding distribution according to four broad categories: (1) monitoring and control, (2) land tenure regularization, (3) sustainable production, and (4) scientific and technological development [13]. However, to understand the allocation of financial resources in light of the design outlined above, it is necessary to further refine the available information from the Amazon Fund. For this purpose, we have built a project database with detailed information on the beneficiaries, activities and geographies that received financial resources from the Amazon Fund (see Figure S8 in Supplementary Materials).

Our primary data source is the Amazon Fund´s website, as well as its annual activity reports (see Figures S3 and S6 in Supplementary Materials). We collected all data available on all of the 96 projects that received support between 2008 and 2017. This data includes project objectives, beneficiaries, implementing organization, territorial scope, committed and disbursed amounts, and activities conducted, among other information. Websites of project owners provided additional information. To refine the data for providing geographical information, we used the municipality as the entity. In Brazil, municipalities reflect the smallest geographical unit for monitoring deforestation, applying public policies, allocating government resources, and evaluating (see Tables S1 and S2 in Supplementary Materials).

One of the main challenges of generating data at the municipal level is the variation of project target areas, which may involve biomes, river basins, protected areas, or indigenous territories. Based on the available literature, we designed rules to determine the municipalities encompassed by each project (see Figure S5 and Tables S3 and S4 in Supplementary Materials). When project disbursements covered multiple municipalities, we used a weight factor in order to determine the share of financial support that each municipality received (see Figure S7 in Supplementary Materials). After the geographical allocation of financial resources, we further categorized the dataset

by main-component, which reflects the Amazon Fund´s theory of change. As projects may contribute to multiple main-components, we conducted one interview by email with a BNDES manager, the managing organization of the Amazon Fund, who replied with a spreadsheet including data dividing the investments of each Amazon Fund project by main-component. Finally, we further categorized the dataset by activity (also called specific-components). As a main-component can be composed by multiple activities, if more than one activity by main-component was verified, then the amounts were equally divided across them. The assumptions in response at divergences or limitations of data collected are presented at Figure S6 in Supplementary Materials. The final database contains 10,493 lines of information structured by project, location, main-component, and specific-component (see Figures S1, S4 and S8 in Supplementary Materials). The procedures for collecting and interpreting data, and constructing the database, are detailed in the supplements outcomes (see Figure S2 in Supplementary Materials). The Amazon Fund accountability is in Brazilian Reais currency. All financial data were converted from Brazilian reais to US dollars by using the rate for the day they were received, which corresponds with the methodology used for the English publications of the Amazon Fund. To evaluate the additionality of the Brazilian governmental agencies budgets (accountable in Brazilian reais) with the Amazon Fund disbursements, we used an average exchange rate between 2009 and 2017, in order to reduce the effects of exchange rate fluctuation.

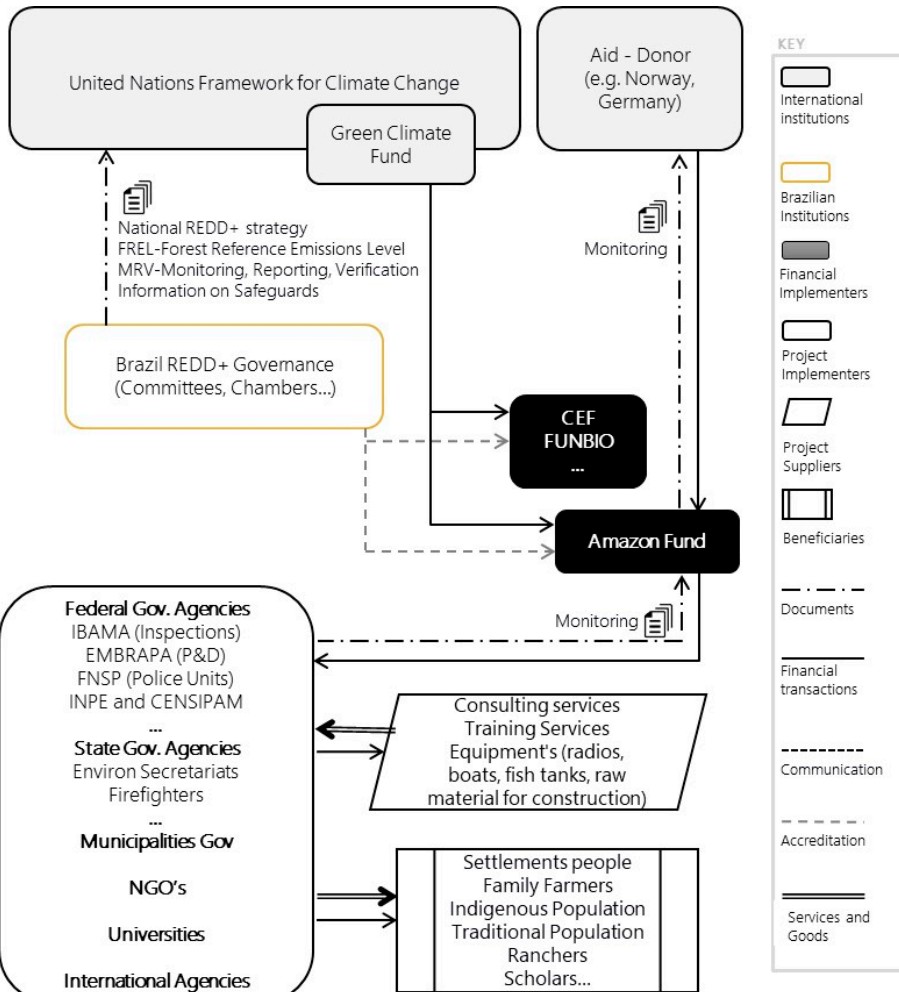

**Figure 1.** The Flows of Amazon Fund. Redd, Reducing Emissions from Deforestation and Forest Degradation; CEF, Caixa Econômica Federal; FUNBIO, Brazilian Biodiversity Fund; IBAMA, Brazilian Institute for the Environment and Renewable Natural Resources; EMBRAPA, Brazilian Agricultural Research Corporation; FNSP, National Police Force; INPE, Brazilian Institute for Space Research; CENSIPAM, Center for the Management of the Amazon Protection System.

## 4. Results: Resource Allocations by the Amazon Fund

Currently, the approval of projects and disbursements are made on the basis of criteria and guidelines updated biannually by COFA. The 2017–2018 document lists 14 minimum requirements that potential projects must meet, some (i.e., items B4, B5, B6, B7, and B14) determining conceptual boundaries of project activities. Projects also must demonstrate coherence with environmental and forest policies, most notably the national Action Plan for the Prevention and Control of Deforestation in the Legal Amazon (PPCDAm), including its manifestations in state governments (PPCDs), and the national policy for Regenerating Native Vegetation (ProVeg) [13]. Projects are also evaluated with respect to coherence with Brazil's National REDD+ Strategy (ENREDD+), which in turn incorporates implementation of PPCDAm and compliance with the Brazilian Forest Code. Finally, projects are expected to be financially additional, i.e., to go beyond existing public environmental budgets and other forms of finance.

The Amazon Fund maintains an open channel for submissions indicating that 80% for the resources should be invested in the Amazon biome (an area that encompasses 40% of the country). In addition to that, the fund also has made public calls aiming at fostering specific activities, such as sustainable production, inclusive value chains and the management of indigenous lands. These calls account for 8.4% of the resources committed by the fund by December 2017. A recent call for forest restauration from 2017 added a spatial priority criteria that provides up to 12 points (from a total of 100) if the project is located in a high priority water basin within a municipality blacklisted as a top priority for deforestation control by the Ministry of Environment [47]. In both the calls and the open submission channel, however, the Amazon Fund adopts largely a passive approach, waiting for project owners to send proposals, rather than actively identifying areas under high risk of deforestation where the impact of the resources would be maximized in terms of deforestation reduction.

### 4.1. Benefit Distribution Across Stakeholders

The distribution of financial commitments across stakeholders shows some variation across years (Figure 2, left panel). In 2017, over 95% of a total of USD 667.3 million went to state governments (USD 256.6 million) or NGOs (USD 241.1 million) or federal governments (USD 140.6 million), with their shares varying considerably per year. Of a total of USD 140.4 million in 2013, about 70% (or USD 102.9 million) went to projects of state governments that received almost no such commitments either two years earlier or two years later. This peak took place as a consequence of a change in the rule of the Amazon Fund that allowed the approval of larger "structural projects", as the implementation of the Rural Environmental Register (CAR). By contrast, commitments to NGOs projects were relatively stable over time, averaging USD 22 million until 2016, though rising to USD 44.5 million in 2017 (implying variation in the NGOs' share). Commitments to federal government projects were also uneven, with slight peaks in 2012 and 2017 (USD 31.7 million, 41.2 million).

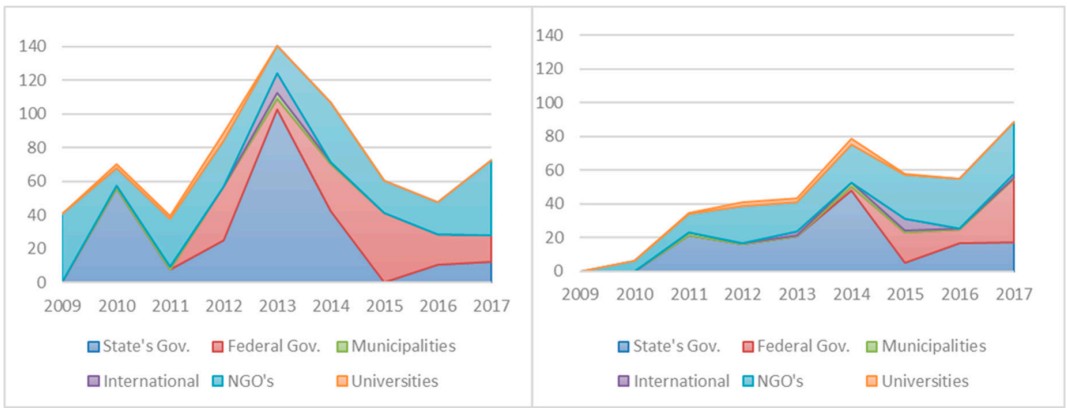

**Figure 2.** Annual committed (**L**) and disbursed (**R**) amounts per stakeholder (in million USD).

However, the ability of different stakeholders to approve projects with the Amazon Fund did not match their implementation capabilities. In the last decade, only USD 405.3 of 667.3 million (i.e., 60.7%) has been transferred to project owners. Average annual disbursements to state governments have hovered between USD 16 and 21 million in most years, with a sudden peak of USD 47.6 million in 2014 and then a sharp drop to USD 4.8 million in 2015. Disbursements to federal government increased exponentially from a small base of only USD 2.4 million even in 2014 to USD 37.7 million in 2017. Finally, disbursements to NGOs steadily increased from USD 6.4 million in 2010 to USD 30.7 million in 2017. From these three groups of beneficiaries, the Federal Government has been demonstrated the largest implementation gap, starting with a very low implementation rate and reaching the execution of only 47% of the committed values by 2017. This was followed by the State Governments, whose spending rates stayed below 50%. Municipalities, Universities, and NGOs, in contrast, presented a better implementation capacity, being able to invest most of the resources obtained from the Fund.

To understand these variations in disbursements, we must also consider the characteristics of the projects supported by the Fund. Federal government projects, for instance, were concentrated within eight projects involving six recipient agencies. Of the total amounts in this category, USD 64.3 million (i.e., 47.2%) went to organizations that develop satellite-based monitoring systems and provide information on deforestation trends, namely INPE and CENSIPAM. Another USD 35.9 million (i.e., 26.7%) went to organizations responsible for enforcing environmental laws and policies, namely IBAMA and FNSP. The remaining USD 40.5 million (i.e., 25.9%) went to EMBRAPA units to disseminate knowledge about sustainable production and the recovery of degraded areas throughout Brazil, and to the Brazilian Forest Service (SFB) for the collection of information aiming to increase the forest data available (see Section 4.3). While the IBAMA manage to invest 17.5% of the funds received, by 2017, INPE and CENSIPAM used only 58.6%, implying that the development of a radar-based monitoring system is lagging behind schedule.

The committed and disbursed peaks for state government projects in 2013 and 2014 (Figure 3) correspond with contextual factors as well, including a surge in state government projects toward development and implementation of the Rural Environmental Register (CAR). CAR is a federal policy instrument introduced in 2012, with the adoption of the new Forest Code (law 12.651/2012), to enhance law enforcement capacity. However, despite the federal law and a centralized national system, the registers must be executed at state or municipal level (art 29, §1). CAR implementation has, therefore, become a major concern for state governments, especially after the system went live in 2014 [48]. This can be seen in both spending and appeals to the Amazon Fund [13]. Within the 13 states that have approved projects, 85% of disbursements went to seven of the nine inside the Amazon Biome.

The linear increase in disbursements to NGOs reflects yet another set of contextual factors, in this case related to Amazon Fund process adjustments over time. Disbursements to projects were slow, to start, due to rigid assessment procedures intended to show professionalism; in the eyes of donor organizations and BNDES management, that slowness also reflected some lack of understanding of project owners [13,19]. Minutes of COFA meetings indicate that, in response to these challenges, the Amazon Fund adopted a number of measures in order to facilitate and accelerate the disbursement process, including public calls for submitting project proposals. While the consequences of these responses are reflected in the linear increase in approved projects and disbursements to NGOs, 80% of the financial resources were concentrated in half of the NGOs that received support from the Amazon Fund. While the Amazon Fund does include distributional equity amongst its performance criteria, this concentration reveals that usually only high-capacity and professional civil society organizations, such as FAS, IMAZON, and TNC, are able to access the fund (see Figure S9 in Supplementary Materials).

In addition to exposing the implementation capability of different governmental agencies, a comparison between the disbursement of the Amazon Fund with the yearly government budget also reveals the ability of the Fund to foster additional actions. One of the key principles of the first donation contract signed in 2008 between Norway and Brazil was the warrant that the Amazon Fund

would not replace but supplement tax payer funds [2,20,22,49]. However, it is possible to observe that the increases in disbursements to federal agencies coincided with their decreasing governmental budgets, particularly after 2014 (Figure 4). This suggests the occurrence of a partial substitution for the agency expenditure of taxpayer-funded budgets using the Amazon Fund. For instance, IBAMA's committed budgets to reduce deforestation, combat fires, and conduct environmental inspections have been reducing since 2012, with a strong reduction from USD 50.64 million in 2014 to USD 29.07 million in 2017. These reductions have been partially offset by Amazon Fund disbursements starting in 2015. Similarly, INPE´s budget fell from USD 84.5 million in 2010 to USD 43.63 million in 2017, 2017, and CENSIPAM has also lost more than 70% of its governmental funding from 2009 and 2017. In those three cases, the Amazon Fund played an important role offsetting those budgetary losses from 2015 onwards, in the case of CENSIPAM even outmatching governmental funds. Those trends include rising implementation rates for turning federal commitments into disbursements, which increased from 3.7% in 2014 to 26.8% in 2017.

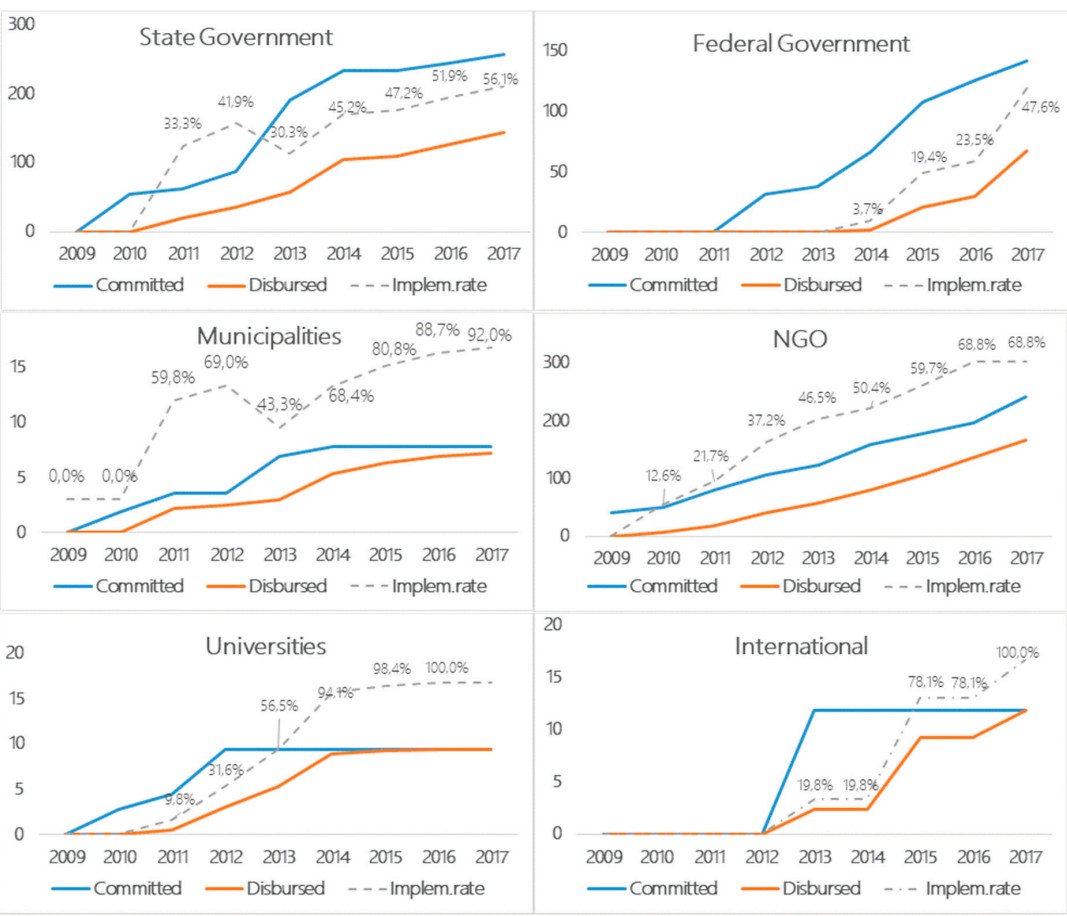

**Figure 3.** Implementation rates as disbursed, divided by committed (consolidated amounts), by Stakeholder.

These observations cannot, by themselves, confirm a direct causal relationship between the increasing financial disbursements from the Amazon Fund and the decreasing budgets of the recipient federal agencies. Furthermore, it should be highlighted that the period following 2015 witnessed one of the worst political, economic, and fiscal crises in Brazil's history. At the same time, however, contextual factors seem to correspond with an interpretation that the forest conservation funding provided through the Amazon Fund lacks in some instances financial additionality, particularly considering the unfavorable political climate for environmental protection [50], the greater flexibility within forest

legislation since 2012 [51], multiple bills for reducing environmental protection during election year 2018, and, as a consequence of all these factors, rising deforestation rates since 2012 [52].

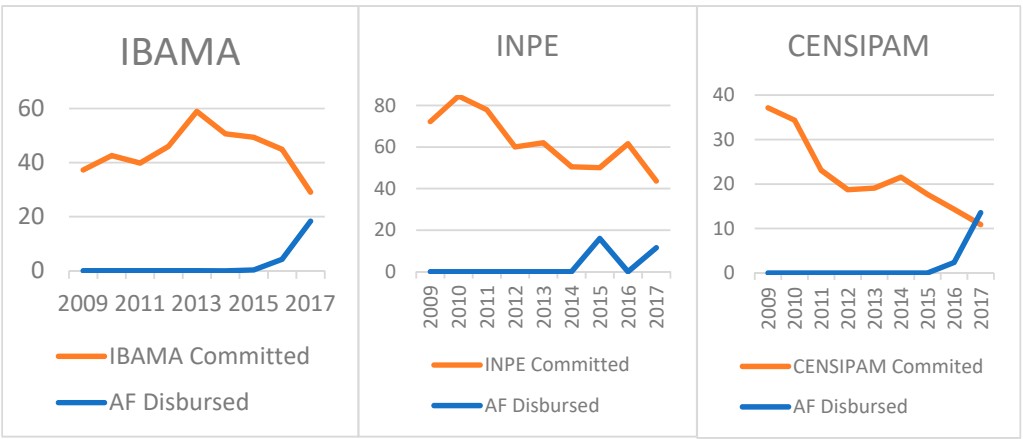

**Figure 4.** Comparison of Federal Committed Budgets with the Amazon Fund disbursements for INPE, IBAMA and CENSIPAM (used average 2009–2017 exchange rate: 2.434). Committed amounts represent the Portuguese term, 'Empenhado', an act that guarantees that there is an amount necessary to pay for an assumed commitment and creates a payment obligation for the government.

*4.2. Geographical Distribution*

Spatially, Amazon Fund allocations display a large concentration (Figure 5a) in 64 municipalities along the (Figure 5a) region stretching from the southeast of Pará towards the western regions in the Mato Grosso, Rondônia and Acre states, municipalities that contain, since 2000, the highest consolidated deforestation rates in Brazil. NGO and state projects explain much of this concentration (Figure 5b,c), whereas federal projects had no significant contribution, mainly due to their nationwide focus (Figure 5c,d). Federal government projects are the most evenly distributed across the landscape, averaging below 26 USD/ha, which could be due to the all-encompassing nature of the geographic information systems (GIS—Geographic Information Systems) and remote sensing activities that these projects tend to promote. At the same time, disbursements to larger federal agencies, such as EMBRAPA, tend to concentrate in eight cities in the Legal Amazon, including Rio Branco, Manaus, Boa Vista and Macapá, where these agencies are located (Figure 5d). Finally, while municipalities benefit indirectly from various types of support, direct support only went to 6 of the 772 municipalities in the Legal Amazon and amounted to only USD 7.8 million. Most of these resources (65.2%) went to the municipal government of Alta Floresta, in northern Mato Grosso. In addition, the Amazon Fund had also financed research of the state universities of Pará (in Belem) and Amazonas (in Manaus) as well as to the development of satellite-based monitoring systems by INPE in Manaus (Figure 5g).

State government projects are mostly responsible for monitoring and control (Figure 5c), particularly through activities, as the structuring of environmental secretariats, CAR implementation, and training of firefighters (see Section 4.3 for details). State governments that more actively sought the support of Amazon Fund for monitoring and control were Acre, Maranhão, Tocantins, and Rondônia. Particularly, Acre has a strong presence in investments in sustainable production, spread throughout its territory (Figure 5e,f). The distribution of resources also portrays low intensity towards Land Tenure Regularization activities, independent of the region or stakeholder (Figure 5h), However, the Amazon Fund allocations did not systematically privilege the municipalities that showed the recent highest deforestation rates. For instance, from the 10 municipalities with the highest deforestation rates in 2017, only two were amongst the top 100 receiving per/Ha, considering the 775 municipalities from Legal Amazon. Furthermore, the support from the Amazon Fund tend to arrive in a context in which clearings have already been reduced substantially due to other factors or the depletion of forests (see Table S5 and Figure S10 in Supplementary Materials). This spatial pattern of project distribution

confirms the apparent lack of strategy of the Amazon Fund, as a consequence of a largely passive approach that waits for proposals rather than actively seeking opportunities for fostering projects in areas with high deforestation risk.

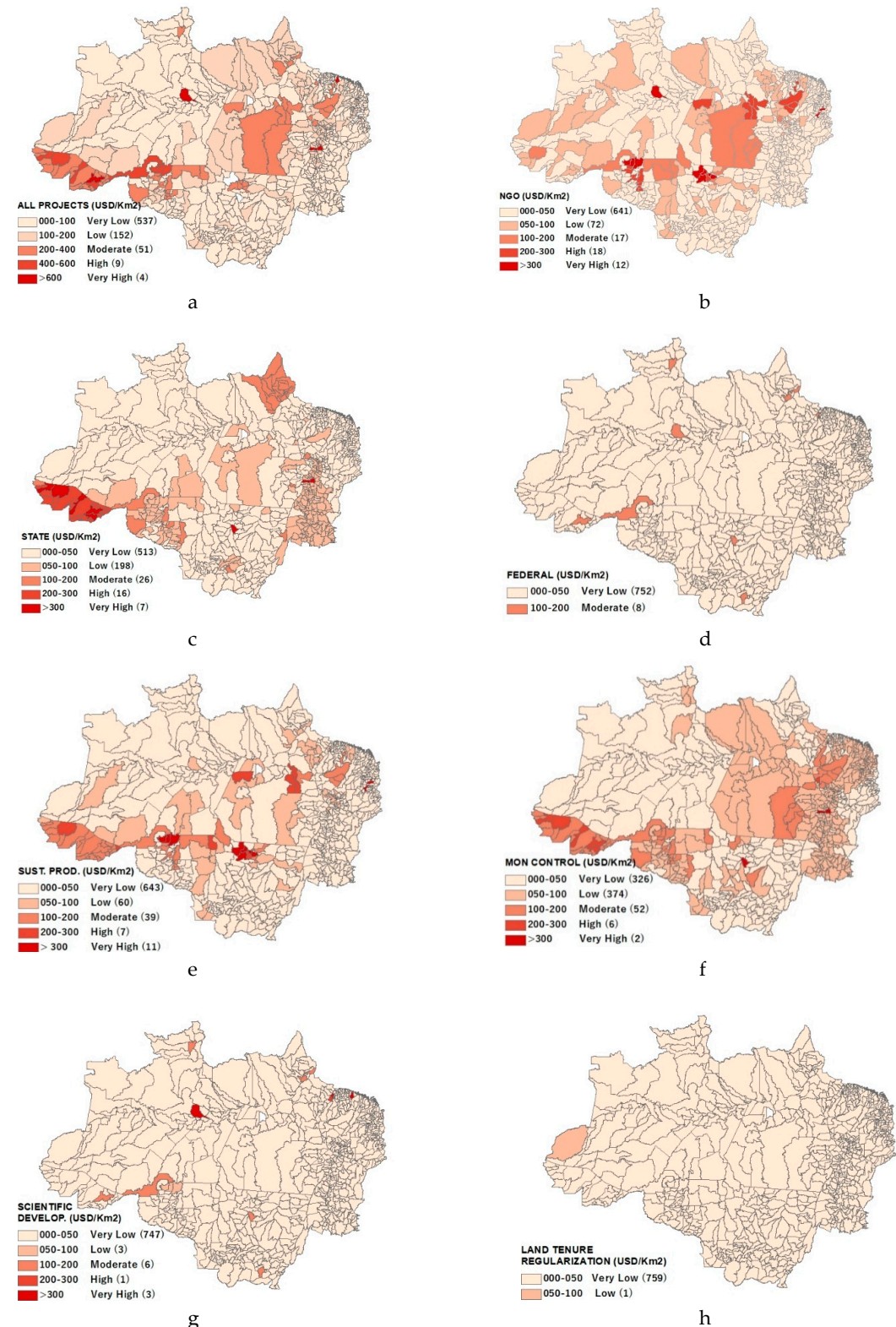

**Figure 5.** Spatial distribution of Amazon Fund investments per municipality by Stakeholder and by main-component.

### 4.3. Distribution Across Activities

Almost half of the total commitments (USD 667.3 million) has gone to monitoring and control (USD 326.7 million), while one third (USD 201.9 million) has gone to sustainable production (see Figure 6 and Table 1). The latter category has been relatively steady over time, as have the small land tenure commitments. By contrast, the large investment monitoring and control have been uneven over time: starting slow with an average of USD 20.3 million in the first four years, peaking in 2013 at USD 94.0 million, and then settling at an average of USD 30.6 million from 2015 on (Figure 6, left panel). Finally, nearly all commitments for scientific and technological development occurred in 2012 (USD 40.7 million).

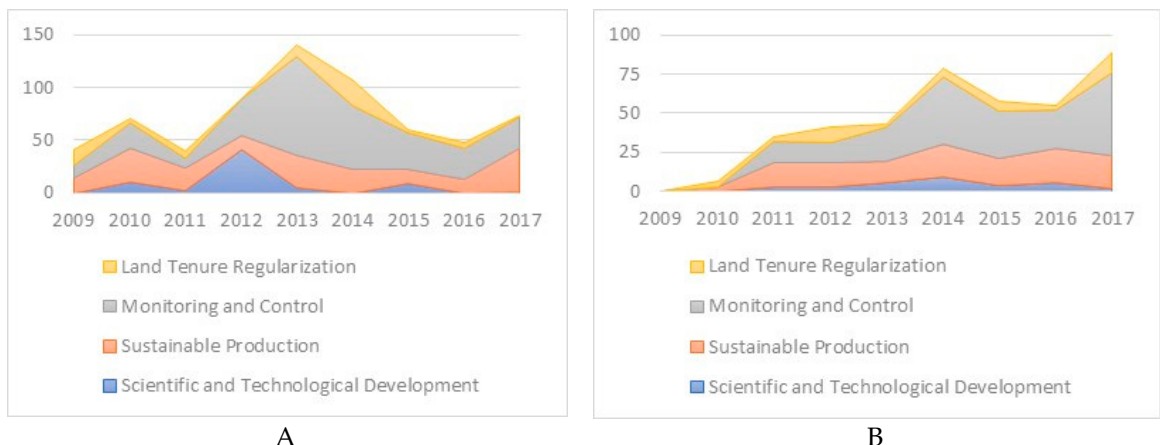

**Figure 6.** Annual committed (**A**) and disbursed (**B**) amounts per main-component (in millions USD).

Although slightly slower than noted above, actual disbursements to individual projects have corresponded to commitments, with most disbursements going to monitoring and control (49.6%) and sustainable production (31.9%). Monitoring and Control was responsible for most of the variation (see right graph of Figure 6), peaking in 2014 (USD 43.1 million) and 2017 (USD 53.5 million). Notably, disbursements for scientific and technological development have never gotten much traction, slightly peaking only in 2013 and 2014, and also presents the lowest implementation rate up to 2017 (Figure 7).

Monitoring and control efforts involved mostly state and federal government projects (USD 187.1 million and USD 100.1 million, respectively). It was the only category, though, that included the unique international project supported by the Amazon Fund, aiming to help develop the capacity to monitor deforestation in eight neighboring countries that also contain the Amazon biome (USD 11.8 million). However, most of the monitoring and control investments (USD 113.0 million) was allocated to CAR implementation. A large share of the funds provided for this activity (USD 102.5 million) was used by state governments to acquire equipment (GPS, computers, software) and provide training for effective processing of CAR proposals. Another share (USD 52 million) was invested in the capacity-building of environmental secretariats for CAR implementation and other environmental policies, including the creation of municipal secretariats, the acquisition of cars and buildings, the hiring of employees and training in-monitoring deforestation, landscape analysis, sustainable supply chains, and measurement. In addition, some resources were used to promote CAR among landowners and to provide georeferencing services for landowners. A small amount went to development of a state system for granting environmental licensing to new businesses and companies. Therefore, in total, 18% of the resources committed by the fund have been invested in the implementation of CAR.

**Table 1.** Distribution of project approvals to Amazon Fund projects (USD). CAR, Rural Environmental Register.

| Activities | State Government | Federal Government | Municipal Government | International | NGOs | Univ. | Total |
|---|---|---|---|---|---|---|---|
| Scientific and Technological Development | 4.457.301 | 40.461.961 | | | 13.990.780 | 9.383.341 | 68.293.383 |
| Field collection and data inventory (Forest, Socioeconomic, Biodiversity, Maps) | 1.771.039 | 31.709.135 | | | 366.095 | | 33.846.268 |
| Disseminate Environmental Education (Museum) | | | | | 5.818.209 | | 5.818.209 |
| Development of New Forest Products | | | | | | 732.695 | 732.695 |
| Develop environmental diagnoses and shared management tools, edit bulletins and publications | | | | | 1.693.133 | 4.736.591 | 6.429.724 |
| Investment in research infrastructure (Laboratories, equipment, facilities, universities) | 1.771.039 | | | | 1.263.966 | 3.914.055 | 6.949.059 |
| Research on the production of native seedlings and techniques for reforestation of degraded areas, development of Demonstration Units (pilots) to disseminate knowledge * | 915.224 | 8.752.827 | | | 4.849.377 | | 14.517.427 |
| Sustainable Production Activities | 41.186.376 | | 5.984.174 | | 154.736.705 | | 201.907.255 |
| Economic Activities for Sustainable Forest Use and Recovery of Degraded Areas | 41.186.376 | | 5.984.174 | | 154.736.705 | | 201.907.255 |
| Monitoring and Control | 187.105.638 | 100.146.294 | 1.788.272 | 11.791.988 | 25.845.426 | | 326.677.619 |
| Structuring and strengthening of State and Municipal Environment Secretariats (Acquire infrastructure, training in Monitoring deforestation, Landscape Analysis, Sustainable Chain and Recovery Measure techniques) | 52.018.486 | | 1.376.210 | | 14.254.668 | | 58.656.955 |
| Inspections, Enforcement and Environmental Police | | 29.571.660 | | | | | 29.571.660 |
| Combat Forest Fires (States–Firefighters/Federal–GIS and Satellites) | 32.543.336 | 6.282.451 | | | | | 38.825.788 |
| Regularize the environmental situation or/and implement CAR | 102.543.816 | | 412.062 | | 11.590.759 | | 113.007.430 |
| Improve Deforestation Monitoring System (GIS and Satellites) ** | | 64.292.183 | | 11.791.988 | | | 76.084.171 |
| Land tenure regularization | 23.829.953 | | 62.995 | | 46.552.443 | | 70.445.392 |
| Land Regularization of Small and Middle size properties (Tenure, Deeds) | 1.141.031 | | | | 3.219.703 | | 4.360.735 |
| Territorial and Ecological Zoning, strengthening and empowerment of PA and IT Management | 22.688.922 | | 62.995 | | 43.332.740 | | 66.084.657 |
| Total | 256.579.269 | 140.608.255 | 7.835.441 | 11.791.988 | 241.125.355 | 9.383.341 | 667.323.649 |

** e.g., improving software; improving the services for receiving, distributing and using satellite images produced; map and make available tools to shape changes in the use of land; improving methods to estimate biomass and emissions; make available a solution for storing and processing a large volume of geospatial data, called "Brazil Data Cube", between others.

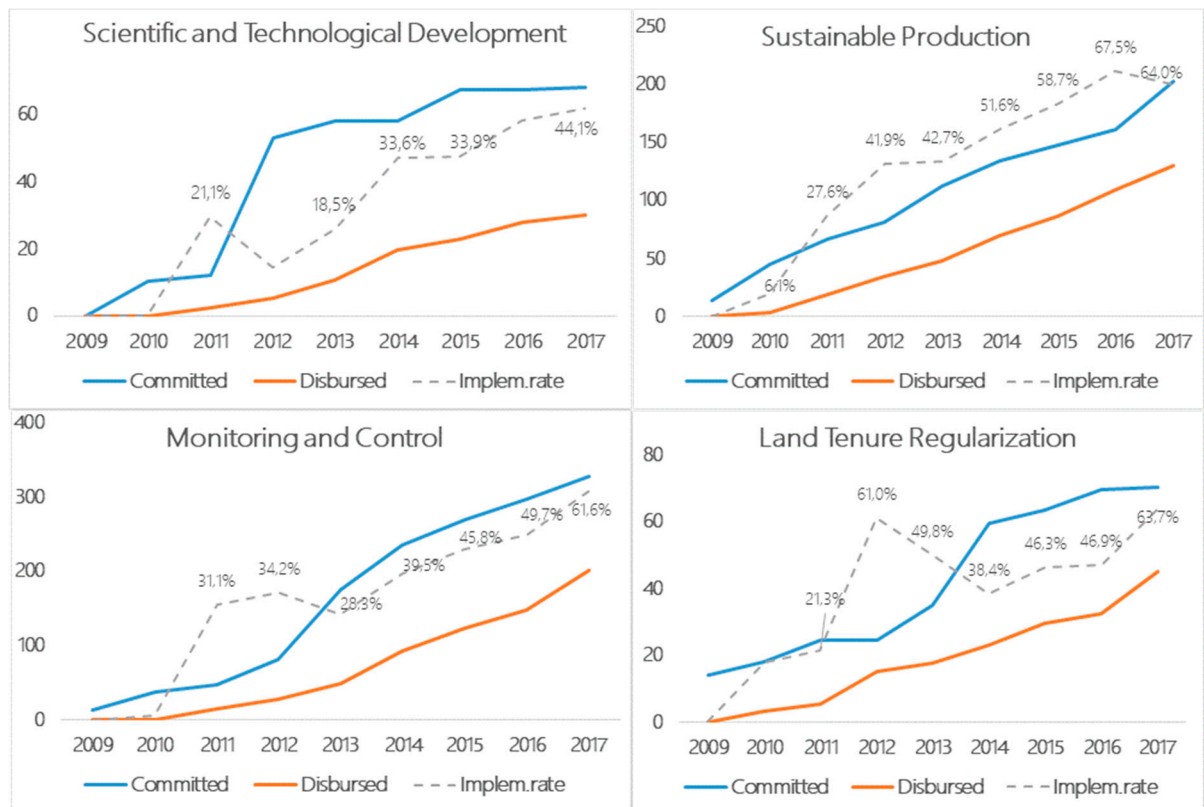

**Figure 7.** Implementation rates as disbursed divided by committed (consolidated amounts), by main-component.

Monitoring activities that were exclusively promoted by federal government organizations involved the improvement of satellite-based monitoring systems for fighting deforestation (PRODES—Annual Deforestation System and DETER—Real Time Deforestation System, USD 76.1 million) and forest fires (PREVFOGO-Fire Prevention System, USD 6.3 million). State governments also invested in forest fire combat (USD 32.5 million), but emphasized control activities (e.g., the creation of firefighter units), rather than monitoring activities. Other investments by federal government organizations targeted the strengthening of law enforcement (USD 29.6 million) in two projects by IBAMA and FNSP; this funding was mostly spent on the acquisition of vehicles, helicopters, equipment, and buildings. While NGOs received much financial support from the Amazon Fund (USD 241.1 million), their support for monitoring and control activities was relatively small (USD 11.6 million) and only involved CAR implementation.

In the category of sustainable production, resources mostly went to NGOs (USD 154.7 million) and state government organizations (USD 42.1 million) (see Table 1). Nearly all state government investments went to the promotion of sustainable forest activities, the acquisition of equipment (tanks, driers, processing units' machines, warehouses), and the provision of professional training and technical assistance (in pisciculture and aquaculture, nut and Açaí extraction, pasture management, as well as forestry and agroforestry systems). This result suggests that the social benefits from the Amazon Fund in terms of rural poverty reduction and sustainable farming were carried out mostly by NGOs and state governments.

Investments in regularizing land tenure, notably spending on territorial zoning and protected-area management and indigenous lands, came almost exclusively from state governments (USD 23.8 million) and NGOs (USD 46.6 million). This investment provides indirect benefits for indigenous peoples, quilombos (descendants from fugitive slaves), riverine people, smallholders, and settlements. No such investments were federal. Federal governments did invest substantially in scientific and technological

development, which involved field data collection by the Brazilian Forest Service (SFB) for building the National Forest Inventory (USD 31.7 million).

Universities, by contrast, invested the most financial resources in scientific research (USD 4.7 million) and development of the research infrastructure (USD 3.9 million). For instance, one project from the Federal University of Pará conducted research for the development of new products from bioactive compounds of plants typical of the Amazon Biome (USD 0.7 million), and invested in the development of new forest products, such as herbal medicines, cosmetics, and food products, among others. Natura, a private cosmetics company from Brazil, announced in 2016 an investment of more than USD 70 million in biodiversity inputs as part of its Amazon Program that aims to develop a new line of products with origins in Amazon Biodiversity.

## 5. Amazon Fund Design Choices and Effectiveness

The findings of our analysis of the recipient projects in the Brazilian Amazon Fund reflect a broad variety of stakeholders and activities. Following the categorization of Luttrell, Loft, Fernanda Gebara, Kweka, Brockhaus, Angelsen, and Sunderlin [27], the recipient projects of the financial resources from the Amazon Fund often involve the largely indirect contributions of effective facilitators, legal rights holders, cost-incurring groups, forest stewards, or poor communities. Moreover, the Amazon Fund's financial resources were channeled towards the direct and structural drivers of deforestation, but this distribution was not proportional to the importance of addressing these drivers, as argued by some scholars (e.g., [32]). Investment patterns tend to reflect specific relations between specific stakeholder groups and project activities. Although activities also vary considerably, there are some general patterns. Federal government organizations tend to invest in development of monitoring systems (45.7%) and inventory data (22.6%), which reflects a main concern with gaining control over deforestation dynamics. State government organizations tend to invest mostly in CAR implementation (40.1%) and capacity-building for state and municipal organizations (20.3%), thereby incurring many of the costs of federal policies. Finally, investments by NGOs have mainly benefited local communities who aim to adopt sustainable production activities (64.2%), but NGOs have also supported (more than federal or state government organizations) land tenure regularization projects (19.3%).

The geographical distribution of financial resources seemed to follow a more focused rationale. We found that many project organizations were located in municipalities with the highest consolidated deforestation rater of Brazil. For instance, NGO projects for territorial and ecological zoning, strengthening of PA and IT management, as well sustainable production, represent 30% of the total disbursements from the Amazon Fund and were largely located in this region. Disbursements from the Amazon Fund to the three main recipient categories have generally benefited municipalities located in areas where deforestation threats are highest [53]. This observation only partially corresponds with the findings by Wolosin, Breitfeller, and Schaap [10], as we found no evidence of substantial contributions to areas with high tree cover, which are more commonly found in remote areas of the Amazon biome [35].

Within the pre-established main-components of the Amazon Fund, we also found variation in the activities that compose these categories. For instance, while most financial resources were channeled to the strengthening of monitoring and control activities by federal and state governments (USD 287.2 million), their investments have focused on monitoring activities like satellite imaging (USD 70.6 million) and CAR implementation (USD 102.5 million). This result contrasts with the substantially smaller investments in control activities like combating forest fires (USD 32.5 million) or law enforcement (USD 29.6 million). This trend is representative of the broader resource allocation within the monitoring and control category. Similarly, investments in land regularization were mainly directed at indigenous territories and protected areas (USD 66.0 million), whereas smallholders (USD 4.3 million) received much less support.

Based on our findings on the variations in financial resource distribution, we argue that the project owners impose a substantial influence on the nature of activities that forest conservation funding

ultimately supports. The current approach adopted by the fund incentivize project submissions in activities and geographies where they may be most successful in reducing deforestation had a limited effect. Corresponding with the study by Weatherley-Singh and Gupta [32], for example, the Amazon Fund restricts financial resource allocation to the four main-components of its theory of change, while not addressing alternative factors, such as the impacts of cattle ranching, road construction, international demand for agricultural products, or corruption. However, any project proposal that adheres to the project quality criteria and guidelines of the Amazon Fund [13] may become eligible for financial support. In other words, the Amazon Fund takes a more passive stance towards resource allocation after the criteria and guidelines are in place. This view accounts for the great variety of stakeholders, activities, and geographies, as described above, since each stakeholder category seems to prefer a different investment strategy. Such behavior may ultimately undermine the effectiveness of conservation funding provided by Norwegian and German donor organizations, at least in terms of emissions reductions.

As already argued in Section 2, the Amazon Fund´s theory of change is generally geared towards deforestation reduction, but the design choices of individual projects are primarily directed at contributing to one or more main components. The evaluation of a completed project in northern Mato Grosso [46], for instance, indicates that the project geared its intervention logic upon its contribution to the main-components "sustainable development" and "monitoring and control", and stated that the main contribution to emissions reductions came from "the restoration of native vegetation and pastures and the planting of native species in permanent protection areas". The extent to which such projects achieved emissions reductions was not stated in the report and would admittedly be a complex methodological endeavor. The leeway that projects have in contributing to these main-components, although important for attracting project proposals, accounts (at least partially) for the imbalanced allocation of financial resources discussed above and may, to some extent, undermine the Amazon Fund's contribution to deforestation reduction.

It is important to note that this undermining of the Amazon Fund´s overall contribution is by no means intentional. At the same time, there are also indications that some projects require a more in-depth evaluation and a longitudinal approach in order to observe their outcomes come to fruition. Particularly but not exclusively, projects from governmental organizations are under greater pressure from critical considerations of their contribution to emissions reductions. One may argue that investments in CAR implementation, for example, support more structural improvements of a nation-wide instrument to enhance monitoring capacity, but some studies point out that it is still unclear whether and to what extent this instrument, indeed, contributes to reducing deforestation [48,54]. In addition, our analysis indicates that federal government organizations (i.e., CENSIPAM, INPE and IBAMA) tend to lack financial additionality. Particularly, the substitutive nature of the Amazon Fund financial resources of IBAMA projects is worrying, because these investments often involve more direct contributions to reducing deforestation, most notably the enhancement of (the capacity for) environmental inspections and fire combat. While the lack of funding for law enforcement may have led to an even higher spike on deforestation rates, a country with a mature enough environmental governance should be able to grant a stable source of public funding by giving priority to this agenda.

## 6. Conclusions

Our analysis also helps to understand why empirical studies seem ambiguous about the effectiveness of forest conservation funding. As explained in Section 3, BNDES' approach to distributing financial resources from the Amazon Fund to individual projects occurs based on the evaluation of project proposals based on the funds widely encompassing guidelines rather than a strategic selection of projects based on the need to reduce deforestation in areas under threat. As a consequence, our findings show that disbursements by the Amazon Fund to individual projects adhere to very diverging theories of change within a broader REDD+ and RBF strategy. The contribution of each individual project for deforestation reduction are complex to be measured and require additional

studies [17,18,22]. Nevertheless, our results suggest that the lack of strategic focus of disbursements may compromise the ability of the fund to obtain further deforestation reductions on the short term.

It is particularly concerning the observation that the resources provided by the Amazon Fund have offset budgetary losses from the Brazilian government in some areas, putting into question the financial additionality of the fund. At the same time, deforestation rates have been on the rise since 2012, the same period during which the fund has started to take place more steadily [19]. It should be emphasized that the fund is not expected to influence deforestation rates for the whole biome, and the lack of additionality in some years can be explained by the economic and fiscal crisis in Brazil. However, these trends taken jointly may weaken the credibility of financial support from the Amazon Fund and other RBF programs on the long term. The sustainable development activities in NGO projects seem to incite less critique, but these projects require much closer scrutiny in order to understand the extent to which they indeed reduce deforestation. Our analysis confirms the argument by Van der Hoff, Rajão, and Leroy [19] that the "demands for demonstrating the results of the Amazon Fund in a scientifically rigorous manner are likely to become an important topic for donor countries".

Alternatively, the Amazon Fund could adopt a more active approach to the allocation of financial resources, for example, by prioritizing a smaller set of activities, with a strong geographical focus. Most importantly, the Amazon Fund should actively identify potential locations and project owners and assist them in constructing high impact proposals. Likewise, the fund should also improve its impact monitoring capabilities and provide incentives to projects that deliver deforestation reductions within the timeframe of the project. This is especially important, as the political climate in Brazil, United States and other countries has become more hostile to environmental interests [52,53,55].

**Supplementary Materials:** The following are available online at http://www.mdpi.com/1999-4907/10/3/272/s1, Figure S1: Model for Database Structuration, Figure S2: Steps to collect the variables, Figure S3: Individual Project Page on Amazon Fund website, Figure S4: Database structured at Level I—Projects, Figure S5: Diagram of rules to determine municipalities encompassed by projects, Figure S6: Project Tree, Figure S7: Municipalities weighted by project, Figure S8: Final Database Structure, Figure S9: Pareto graft for NGO's and State projects (USD left bar and % of committed amounts right side, Figure S10: Deforestation in Legal Amazon, PRODES-INPE (2017), Table S1: Municipalities geospatial information sources, Table S2: Municipalities Data Source, Table S3: Variables included in the main-component level, Table S4: Weight calculations per main-component, Table S5: 10 Municipalities with the higher deforestation rates between 2016 to 2017. PRODES-INPE (2017), Table S6: Research assumptions in response at divergences/limitations of data collection.

**Author Contributions:** Conceptualization, J.C., R.v.d.H. and R.R.; Data curation, J.C.; Formal analysis, R.v.d.H. and R.R.; Investigation, J.C. and R.v.d.H.; Methodology: J.C. and R.R.; Visualization, R.R.; Writing—Review & Editing, J.C., R.v.d.H. and R.R.

**Funding:** This research was supported by the Universidade Federal de Minas Gerais, through the Pro-Rectory of Research-PRPq/UFMG. Also was financed by CAPES–Brazilian Federal Agency for Support and Evaluation of Graduate Education within the Ministry of Education of Brazil.

**Acknowledgments:** We thank Alexander Pfaff from Duke University for the inspiration and academic support, and Centro de Inteligência Territorial (CIT), PRPQ/UFMG and CAPES for providing funding for this research.

**Conflicts of Interest:** The authors declare no conflict of interest.

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
