# Peer review of "Amazon Fund 10 Years Later: Lessons from the World’s Largest REDD+ Program"

_forests, doi:10.3390/f10030272_

Round 1

Reviewer 1 Report

See attached

Author Response

Thank you for your rich recommendations. In this round of revision, we rewritten large parts of the paper in order to eliminate redundancies, and we used terminology properly and avoid English mistakes. We also tried attend your recommendations and pointed our considerations, as follow in the attached document.

Reviewer 2 Report

Thank you for the opportunity to review this manuscript. I find the data presented to be quite enlightening, and I think it is now pretty clear what the authors' contribution to the literature is. I have a few very minor thoughts that they might consider in revisions.

The largest issue is that I still think the principal-agent perspective is being treated as a bit of a strawperson; while it may be the case that the basic model as it is discussed in the the REDD+ context is a two-person principal-agent model, the approach allows for more complex models, as well, with longer chains of principals and agents. This happens, for example, in the literature applying principal-agent approaches to international organizations. It might be a good idea to nuance this discussion a bit, as this can then be returned to in the discussion and conclusion, as I think the manuscript does speak to the issue of lengthening chains of principals and agents in a way that is consistent with some of these discussions and could make the results even more relevant for the broader literature.

In addition, there are the following typos that I noticed:

Line 238 needs to be indented

Line 281 - Distribution is misspelled

Line 293 - have -> has

The addition of absolute numbers in Figure 3 makes for some very messy graphics.

Line 352 - this sentence is ungrammatical; perhaps there is a word missing between NGOs and high-capacity - or a comma?

Line 386 - distribution is misspelled

Line 408 - odd to say these companies are covered by the Amazon biome; rather, better to say they contain it

Author Response

Thank you for review our manuscript.

Indeed, now in this revision we have opted to nuancing the discussion in the paper if it is a basic or a two-person model, and we prioritized highlight the issues under the principal-agent perspective.

We have rewritten large parts of the paper in order to eliminate redundancies, use terminology properly and avoid English mistakes.
